# Biosynthetic Mechanisms of Plant Chlorogenic Acid from a Microbiological Perspective

**DOI:** 10.3390/microorganisms13051114

**Published:** 2025-05-13

**Authors:** Jiasi Zhong, Qingsong Ran, Yanfeng Han, Longzhan Gan, Chunbo Dong

**Affiliations:** Institute of Fungus Resources, Department of Ecology/Key Laboratory of Plant Resource Conservation and Germplasm Innovation in Mountainous Region (Ministry of Education), College of Life Sciences, Guizhou University, Guiyang 550025, China; 17390409701@163.com (J.Z.); 15185968097@163.com (Q.R.); yfhan@gzu.edu.cn (Y.H.); lzgan@gzu.edu.cn (L.G.)

**Keywords:** chlorogenic acid, microbial ecology, plant symbiosis, synthetic microbial consortia, genetic modification

## Abstract

Chlorogenic acid (CGA), a phenolic compound with diverse bioactivities, plays a crucial role in plant defense mechanisms and has significant therapeutic potential in human inflammatory and cardiovascular diseases. The biosynthesis and accumulation of CGA in plants result from a complex interplay between internal factors (e.g., hormones, enzymes, and genes) and external factors (e.g., microbial interactions, drought, and temperature fluctuations). This review systematically investigates the influence of microbes on internal regulatory factors governing CGA biosynthesis in plants. CGA is synthesized through four distinct metabolic pathways, with hormones, enzymes, and genes as key regulators. Notably, microbes enhance CGA biosynthesis by improving plant nutrient uptake, supplying essential hormones, regulating the expression of related enzymes and genes, and the interaction between bacteria and fungi. In addition, our review summarizes the challenges currently present in the research and proposes a series of innovative strategies. These include in-depth investigations into the molecular mechanisms of microbial regulation of plant gene expression, gene editing, development of microbial inoculants, construction of synthetic microbial communities, and exogenous application of plant hormones.

## 1. Introduction

In recent decades, phenolic compounds have been recognized for their significant medicinal value and therapeutic potential [1]. Chlorogenic acid (CGA; also known as 3-caffeoylquinic acid, 3-CQA) is a phenolic compound classified within the phenylpropanoid family. It is naturally synthesized through the esterification of the carboxyl group in trans-cinnamic acids—predominantly caffeic acid, *p*-coumaric acid, and ferulic acid [2]—and the hydroxyl group of quinic acid [3]. The molecular structure of CGA is distinguished by three labile chemical groups: an ester bond, an unsaturated double bond, and polyphenolic hydroxyl groups [4]. CGA exists in several isomeric forms, including neochlorogenic acid (5-CQA), cryptochlorogenic acid (4-CQA), and coffeoyl quinic acid (3-CQA) [5].

CGA is prevalent in various plants, including *Eucommia ulmoides*, *Camellia sinensis*, and *Lonicera japonica* [6]. CGA confers resistance to herbivory and pathogen attacks and enhances tolerance to environmental stresses such as high temperature, salinity, and intense light [7]. Further, CGA possesses a wide array of pharmacological properties, including anti-inflammatory, antitumor, antibacterial, and immunomodulatory effects [8]. As a secondary metabolite, CGA biosynthesis and accumulation are regulated by endogenous and exogenous factors [9]. Endogenous factors include genetic background [10], developmental stage, tissue specificity, and metabolic pathways, whereas exogenous factors include light, temperature, soil moisture, soil fertility, salinity, and microbial interactions [9].

Microbes are widely recognized as essential regulators of plant secondary metabolism, playing a critical role in plant growth and the synthesis of secondary metabolites [11]. Plant growth-promoting bacteria (PGPB), for example, promote plant growth and secondary metabolism by improving soil nutrient uptake through processes such as biological nitrogen fixation, phosphate solubilization, siderophore production, and 1-aminocyclopropane-1-carboxylate (ACC) deaminase activity, thereby facilitating the acquisition of nitrogen, phosphorus, and essential minerals [12,13]. Moreover, PGPB can synthesize plant hormones, including cytokinins (CK), indole-3-acetic acid (IAA), and gibberellins (GA), all of which influence plant growth and metabolism [14]. These functions are mediated through the regulation of gene expression, involving both the activation of specific microbial genes and the modulation of plant gene expression, ultimately promoting secondary metabolite synthesis. For instance, *Bacillus subtilis*, *B. licheniformis*, *Burkholderia*, and *Acinetobacter pittii* increase the production of plant terpenoids [15], while *B. halodurans* SCCPVE07 promotes the synthesis of phenolic compounds [16]. Similarly, *Azotobacter chroococcum Beijerinck* and *Pseudomonas putida* boost the production of plant alkaloids [17]. Despite the well-documented role of microbes in promoting the accumulation of plant secondary metabolites such as terpenoids, phenolics, and alkaloids, research into the synthesis and accumulation of the specific secondary metabolite, CGA, is lacking. Specifically, there is a paucity of systematic studies into microbial regulation of CGA synthesis. This review aims to address these gaps by pursuing two main objectives: (1) elucidating the pathways, hormones, enzymes, and genes involved in CGA biosynthesis in plants; and (2) clarifying the mechanisms by which microbes influence CGA synthesis. Finally, we summarize the current challenges surrounding the study of microbial impacts on CGA and other secondary metabolites and propose new directions for future research.

## 2. Chlorogenic Acid Functions

### 2.1. Biological Functions of Chlorogenic Acid

In natural environments, plant growth and development are challenged by a variety of biotic and abiotic stresses. To mitigate the adverse effects of these stressors, plants synthesize CGA, enhancing their natural resistance and antioxidant capacity, thereby reducing the damage inflicted by external stresses (Figure 1).

Intracellular CGA levels in plants have been shown to significantly increase in response to biotic stresses, including herbivory, insect predation, and pathogenic fungal infection, thereby exhibiting notable anti-herbivore and antimicrobial activities [18,19]. For example, infestation by the *Cylas formicarius* results in elevated CGA content in *Ipomoea batatas* leaves, which confers an antagonistic effect against the weevil [20]. Similarly, feeding by the *Spodoptera litura* on *Solanum lycopersicum* induces a considerable increase in CGA levels in both mature and young leaves [21]. In addition to its role in insect resistance, CGA demonstrates significant antimicrobial activity during plant defense against pathogenic fungi. Gray mold, caused by *Botrytis cinerea*, severely impacts the growth and quality of fruit in peach trees. However, CGA has been shown to effectively inhibit the growth of *B. cinerea* [22]. These findings underscore the multifaceted defensive properties of CGA against both insect pests and pathogenic fungi, highlighting its potential as a valuable tool in plant disease management strategies.

Beyond its defensive functions, CGA serves as an antioxidant by efficiently scavenging reactive oxygen species (ROS) in plants. Under abiotic stresses such as drought, high temperature, salinity, and intense light, plants increase ROS expression, including superoxide radicals, hydrogen peroxide, and hydroxyl radicals [23]. Excessive ROS can induce oxidative damage to cell membranes, proteins, RNA, and DNA, ultimately causing cell aging and death in severe cases [24]. CGA effectively neutralizes these ROS, thereby reducing oxidative damage to plants. For example, the exogenous application of CGA has been shown to slow the rate of chlorophyll degradation, mitigate membrane damage and lipid peroxidation, and enhance antioxidant enzyme activity in apple slices, thus alleviating oxidative stress responses in vitro [23]. Similarly, *Triticum aestivum* and *Aegilops cylindrica* produce significant amounts of superoxide radicals and hydrogen peroxide under salt stress. During this process, the levels of CGA and other phenolic compounds increase, exerting an inhibitory effect on ROS [25].

### 2.2. Pharmacological Functions of Chlorogenic Acid

CGA exerts multidimensional therapeutic effects across various organs, driven by its antioxidant and anti-inflammatory properties [3]. The antioxidant capacity of CGA is attributed to the hydrogen-donating ability of its multiple phenolic hydroxyl groups, which neutralize free radicals generated by metabolism in humans. The stable phenoxyl radicals formed upon CGA oxidation further protect human cells from oxidative damage [26]. In its anti-inflammatory role, CGA inhibits the synthesis and secretion of pro-inflammatory cytokines, including TNF-α, IL-1β, IL-6, and IL-8, as well as nitric oxide and prostaglandin E2 [4]. Furthermore, CGA regulates key signaling pathways and related factors to mitigate inflammatory diseases. For example, CGA activates the NRF2/HO-1 signaling pathway while inhibiting NF-κB transcription factor signaling, thereby alleviating oxidative stress and inflammatory responses [27]. Additionally, studies by Xiong et al. have demonstrated that CGA suppresses the NF-κB transcription factor in the TNF signaling pathway and activates the MAPK signaling pathway, leading to the downregulation of inflammatory mediators such as TNF-α, MMP9, and PTGS2 [28].

In oncology, CGA has been shown to inhibit the expression of programmed death-ligand 1 (PD-L1) in tumor cells stimulated by IFN-γ. This effect is mediated through suppression of the p-STAT1-IRF1 signaling pathway and increased T-cell activation [29]. In cardiovascular disease models, CGA pretreatment in rats reduces infarct size, diminishes pro-inflammatory cytokine activity, and augments anti-inflammatory and antioxidant enzyme activities, thereby mitigating acute myocardial infarction [30]. In the context of neurodegenerative diseases, CGA and NCGA exert neuroprotective effects via three mechanisms: alleviating oxidative stress, reducing neuronal inflammation, and enhancing the expression of brain-derived neurotrophic factor (BDNF) [31]. In diabetes, CGA regulates carbohydrate and lipid metabolism, providing a potential therapeutic strategy for diabetes and its associated complications [32]. In metabolic disorders, CGA ameliorates hyperuricemia and its associated inflammatory responses and gastrointestinal dysfunction in mice induced by hypoxanthine and potassium oxonate. This effect is achieved through reductions in uric acid, blood urea nitrogen, enzyme activity, and mRNA expression [33].

The intestines are identified as the primary sites for the metabolism and absorption of CGA in the body, with these processes being significantly influenced by the host gut microbiota [34]. Specifically, one-third of dietary CGA is absorbed in the small intestine, where it undergoes biochemical transformations to produce quinic acid, ferulic acid, and caffeic acid. Conversely, the remaining two-thirds of CGA proceed to the large intestine, where they are rapidly degraded by intestinal microbiota [35]. The degradation of CGA in the large intestine was investigated using high-performance liquid chromatography–mass spectrometry (HPLC-MS) and gas chromatography–mass spectrometry (GC-MS), resulting in the identification of 11 degradation products. Among these, dihydrocaffeic acid, dihydroferulic acid, and 3-(3′-hydroxyphenyl)propionic acid were found to be the predominant metabolites, collectively accounting for 75–83% of the total metabolites [36].

## 3. Chlorogenic Acid Production in Plants

CGA, a secondary metabolite known for its antioxidant and anti-inflammatory properties, is synthesized in various plants. To elucidate the biosynthetic mechanisms of CGA, this review systematically examines its biosynthetic pathways, enzymatic activities, hormonal regulation, and gene expression, thereby providing a comprehensive understanding of the routes and regulatory elements involved in CGA biosynthesis.

### 3.1. Chlorogenic Acid Biosynthesis

CGA biosynthesis is initiated through photosynthesis, during which glucose is generated and subsequently converted into phosphoenolpyruvate (PEP) and D-erythrose 4-phosphate via glycolysis and the pentose phosphate pathway (PPP) [37]. These intermediates are then directed into the shikimate pathway to form chorismate, which is further converted to L-phenylalanine through a series of enzymatic reactions. L-phenylalanine serves as the starting material for the phenylpropanoid pathway, where the initial steps—conversion of L-phenylalanine to *p*-coumaroyl-CoA by phenylalanine ammonia-lyase (PAL), cinnamate 4-hydroxylase (C4H), and 4-coumarate-CoA ligase (4CL)—are collectively termed the general phenylpropanoid pathway [38]. The resulting *p*-coumaroyl-CoA represents a branching point for two of the four known CGA biosynthetic pathways. In pathway I, which is predominant in potatoes [39], *p*-coumaroyl-CoA is esterified with quinic acid to form *p*-coumaroylquinic acid, which is then hydroxylated by *p*-coumarate 3-hydroxylase (C3H) to yield CGA.

In eggplants [40], the same pathway utilizes hydroxycinnamoyl-CoA shikimate/quinate hydroxycinnamoyl transferase (HCT) instead of hydroxycinnamoyl-CoA: quinate hydroxycinnamoyl transferase (HQT) for the esterification step. Pathway II, the major route in artichokes and switchgrass, begins with the esterification of *p*-coumaroyl-CoA and shikimic acid by HCT, releasing CoA to form *p*-coumaroyl shikimate [41]. This intermediate is hydroxylated by coumaroyl shikimate 3′-hydroxylase (C3′H) to produce caffeoyl shikimate. Caffeoyl shikimate can either be esterified by caffeoyl shikimate esterase to form caffeic acid or de-esterified by HCT to yield caffeoyl-CoA. The resulting caffeoyl-CoA is then esterified with quinic acid (or shikimic acid) by HQT (or HCT) to form CGA [42]. In the final step of this biosynthetic route, HCT preferentially forms CGA with shikimic acid and caffeoyl-CoA, whereas HQT preferentially forms CGA with quinic acid and caffeoyl-CoA [43] (Figure 2).

Beyond the pathways previously detailed, two additional biosynthetic routes for CGA have been identified in plants. Pathway III initiates with *p*-coumaric acid, which is converted to caffeic acid via esterification by either C4H or C3H. The resulting caffeic acid is subsequently activated to caffeoyl-CoA by 4CL and proceeds through the terminal step of Pathway II to yield CGA [42]. Pathway IV, originating from cinnamic acid, involves the sequential actions of UDP glucose: cinnamate glucosyltransferase (UGCT) and hydroxycinnamoyl glucose: quinate hydroxycinnamoyl transferase (HCGQT) to synthesize CGA [44] (Figure 2).

### 3.2. The Hub of Biosynthetic Pathways—Key Enzymes

In plants, the biosynthesis of both primary and secondary metabolites depends on enzymatic activity. Enzymes play dual roles in these processes: first, they direct the course of biochemical reactions. Within the intricate metabolic network, distinct enzymes catalyze specific reaction steps, channeling substrates through defined metabolic pathways [45]. Second, enzymes regulate reaction rates. In the CGA biosynthetic pathway, PAL, C4H, and 4CL act as key rate-limiting enzymes in the initial steps of synthesis [46]. PAL is crucial for the transition from primary to secondary metabolism, initiating the synthesis of phenylpropanoid-derived secondary metabolites [47]. In *I. batatas*, PAL accumulation correlates significantly with CGA content across different tissues and developmental stages [48]. In *Camptotheca acuminata*, gene cloning and functional identification have confirmed C4H as a pivotal enzyme in CGA biosynthesis [49]. Additionally, analysis of the *E. ulmoides* 4CL gene family has demonstrated that 4CL is vital for CGA production [50].

Beyond PAL, C4H, and 4CL, acyltransferases are essential for CGA biosynthesis, with HQT and HCT being key enzymes involved in CGA formation and hydrolysis [51]. These enzymes are integral to plant growth. For example, induction with methyl jasmonate (MeJA) increases HCT synthesis, resulting in major CGA accumulation in Carthamus tinctorius cells [52]. Additionally, a recent study demonstrated that manipulating HQT expression in *S. lycopersicum* leaves leads to substantial changes in CGA content [53]. Other enzymes, including C3H, C3′H, and UGCT, also play crucial roles in CGA biosynthesis. Collectively, these enzymes act synergistically to regulate CGA levels in plants.

### 3.3. Regulators—Hormones

Plant hormones regulate not only growth and developmental processes but also modulate metabolic pathways, key enzymes, and gene expression, thereby influencing the synthesis of secondary metabolites [54]. These hormones include salicylic acid (SA), jasmonic acid (JA), GA, ethylene (ET), abscisic acid (ABA), IAA, CK, and melatonin (MT) [55]. These hormones have been shown to impact CGA biosynthesis in plants, as discussed in subsequent sections.

SA, a plant hormone derived from isochorismate, acts as an inducer to increase the synthesis of secondary metabolites in plants [56]. SA and its derivative, methyl salicylate, have been shown to activate the activity of PAL, leading to a 55% increase in CGA content in apple leaves [57]. In *I. batatas* shoot tips, SA treatment for 72 h significantly increased CGA content relative to controls [58]. Similarly, application of 100 mg/L SA to grapevines resulted in a marked rise in total phenolic content, with CGA levels showing the most substantial increase [59]. Furthermore, SA upregulates the expression of genes associated with CGA biosynthesis, including *PAL*, *C4H*, and *HQT*, thereby boosting CGA production in *I. batatas* leaves and enhancing resistance to *C. formicarius* infestation [20]. These results demonstrate that SA promotes CGA biosynthesis in plants by activating relevant genes, underscoring its role as a key regulator in plant physiological processes.

JA is integral to plant growth and developmental regulation, and is also key for promoting the differentiation of reproductive organs. JA has been shown to improve CGA synthesis by modulating the expression of genes including *Ib4CL*, *IbHCT*, *IbPAL*, and *IbHQT* in *I. batatas* [20]. Park et al. reported that treating germinating buckwheat with 150 μM JA for 72 h yielded the greatest increase in CGA concentration relative to other hormonal treatments [60]. Similarly, the treatment of peaches with MeJA elevated PAL and 4CL activities, resulting in significant increases in CGA and other phenolic compounds, such as 5-CQA [61].

Beyond SA and JA, additional plant hormones variably influence CGA biosynthesis, having either stimulatory or inhibitory effects. The CGA content in *Hordeum vulgare* seedlings was found to gradually decrease when treated with GA at concentrations ranging from 0.1 to 1.0 mg/L [62]. In green cherry tomatoes soaked in 1.0 mM ABA, a significant accumulation of various phenolic compounds was observed, although the CGA content exhibited an initial increase followed by a decrease during ripening [63]. In a study by Li et al., pitaya fruit soaked in a 1000 μmol L^−1^ ethephon solution for 10 min exhibited activated key enzyme activities in the phenylpropane metabolic pathway and induced phenolic compound synthesis within the fruit, thereby accelerating the formation of CGA, gallic acid, and protocatechuic acid [64]. The callus tissue of *Bidens pilosa* was induced using varying ratios of 2,4-dichlorophenoxyacetic acid (2,4-D, auxin) and benzylaminopurine (BAP, cytokinin). The CGA content in *B. pilosa* callus tissue was significantly elevated when cultured in a medium containing 2,4-D (2 mg/L) and BAP (0.2 mg/L) at a ratio of 10:1 [65]. Collectively, these findings suggest that most plant hormones enhance the expression of genes and the activity of enzymes associated with CGA synthesis, thereby promoting CGA production in plants.

### 3.4. The Manipulators Behind Chlorogenic Acid—Genes

#### 3.4.1. Enzyme Gene

PAL, a key rate-limiting enzyme in the phenylpropane metabolic pathway, plays a vital role in plant growth, development, and adaptation to environmental stress [66]. Initially isolated from *H. vulgare*, the PAL gene has subsequently been identified across diverse organisms, including plants, viruses, algae, and fungi [67]. Typically encoded by a small multigene family [68], the PAL gene family comprises varying numbers of members depending on the species [67]. In research on the CGA biosynthetic pathway in sweet potato, plants transformed with an *IbPAL1* overexpression vector exhibited a 2- to 2.5-fold increase in PAL activity compared to the control group, with *IbPAL1* expression positively correlated with CGA content [69].

C4H, the second key enzyme in the general phenylpropane pathway, belongs to the cytochrome P450 family (*CYP73A*) [70]. Genes encoding C4H have been identified and cloned from various plants, including *Arabidopsis thaliana* [71] and *Brassica napus* [72]. Four distinct C4H genes (*PtreC4H1-1*, *PtreC4H1-2*, *PtreC4H2-1*, and *PtreC4H2-2*) have been identified in Populus tremuloides. Upon cloning the *PtreC4H1-1* and *PtreC4H2-1* genes into *Saccharomyces cerevisiae* for expression, the yeast demonstrated the ability to convert trans-cinnamate to *p*-coumaric acid, thereby confirming the activity of the C4H enzymes encoded by these genes [73]. Furthermore, the *IbC4H* gene is cloned, which encodes the key enzyme C4H in the phenylpropane pathway of sweet potato cv. XZ 3, revealed a significant positive correlation between *IbC4H* expression levels and phenolic acid content. This finding suggests that *IbC4H* may be a critical component of the regulatory mechanism underlying phenolic biosynthesis in sweet potato [74].

In plants, 4CL is present in multiple isoenzyme forms and is encoded by a small gene family that produces proteins with similar functions or structures [75]. The 4CL gene has diverged into two distinct groups in monocots and dicots. In dicots, 4CL genes are divided into two clusters: Type I and Type II, while in monocots, 4CL genes are divided into two clusters: Type III and Type IV [76]. Four 4CL genes (*Mn4CL1*, *Mn4CL2*, *Mn4CL3*, and *Mn4CL4*) were cloned from *Morus atropurpurea* cv. Jialing No. 40. Tissue-specific expression analysis revealed that the expression level of *Mn4CL3* was higher than that of the other genes and correlated with the trend of total flavonoid content during fruit development [77]. In tea plants, two 4CL homologous genes, *Cs4CL1* and *Cs4CL2*, were assigned to Class I and Class II of this gene family, respectively. Enzyme activity assays demonstrated that both recombinant *Cs4CL1* and *Cs4CL2* could convert 4-coumaric acid, ferulic acid, and caffeic acid into their corresponding CoA esters. Meanwhile, overexpression of *Cs4CL1* and *Cs4CL2* in transgenic tobacco seedlings increased the levels of CGA and total lignin [78].

In the realm of plant CGA biosynthesis, the influence of HCT genes on CGA production has been explored across diverse plant species. For example, MeJA activates the expression of the *CtHCT* gene in safflower, resulting in elevated intracellular CGA accumulation [52]. In transgenic *N. tabacum*, expression of the *HCT-45178* gene yields a 54–149% increase in CGA content relative to wild-type plants [79]. Additionally, in apples, the expression levels of *MdHCT1*, *MdHCT2*, *MdHCT4*, *MdHCT5*, and *MdHCT6* genes are positively correlated with CGA content [80].

Investigations into the function of HQT genes have demonstrated that their expression significantly boosts CGA biosynthesis in diverse plant species. For example, expression of the *LmHQT* gene in *N. tabacum* leaves increases CGA content [81]. In artichokes, expression of the *HQT1* gene enhances the production of CGA and cynarin [82]. Moreover, integrated transcriptomic and metabolomic analyses of CGA biosynthesis in *L. japonica* indicate that overexpression of *LmHQT* and *LmHCT* genes results in higher CGA levels [81].

#### 3.4.2. Hormone Biosynthesis and Regulatory Genes

CGA synthesis is regulated by hormones; the biosynthesis of which is, in turn, controlled by genetic factors. For example, the biosynthesis of SA is regulated by genes encoding several key enzymes, including isochorismate synthase (ICS), PAL, avrPphB SUSCEPTIBLE3 (PBS3), enhanced disease susceptibility 1 (EDS1), chorismate mutase, enhanced disease susceptibility 5 (EDS5), and abnormal inflorescence meristem1 (AIM1) [83]. Similarly, JA biosynthesis is influenced by multiple genes encoding enzymes such as phospholipase A (PLA), lipoxygenase (LOX), allene oxide synthase, allene oxide cyclase (AOC), 12-oxophytodienoic acid reductase 3 (OPR3), and acyl-CoA oxidase (ACX) [84]. GA production is also a result of the coordinated action of multiple genes, including copalyl diphosphate synthase (CPS), kaurene synthase, kaurene oxidase, kaurenoic acid oxidase (KAO), 2-oxoglutarate-dependent dioxygenases, gibberellin 20-oxidase (GA20ox), and gibberellin 3-oxidase (GA3ox) [85]. Collectively, the synthesis of each hormone is dependent on the synergistic functioning of multiple genes, and the expression of these genes indirectly modulates the biosynthesis and accumulation of secondary metabolites, including CGA.

#### 3.4.3. Transcription Factors

The *MYB* gene family members are characterized by a highly conserved DNA-binding domain located at the N-terminus, and the resultant MYB transcription factors are implicated in diverse biological processes, including secondary metabolism, hormone signaling, responses to environmental stresses, cell differentiation, and the cell cycle [86]. In *L. macranthoides*, *LmMYB3*, *LmMYB4*, *LmMYB15*, and *LmMYB31* exhibit significant correlations with CGA content. Notably, *LmMYB15* has been shown to specifically bind to and activate the promoter regions of *4CL*, *MYB3*, and *MYB4*, thereby enhancing CGA biosynthesis [87].

The *WRKY* gene family constitutes a major class of transcriptional regulators in plants, controlling downstream transcription through specific binding to W-box cis-elements (TTGACT/C) in the promoter regions of genes associated with growth, development, and stress responses [88]. In *Taxus antungensis*, overexpression of the *TaWRKY14* gene, previously isolated, results in elevated expression of *TaPAL1* and a corresponding increase in CGA concentration [89].

bHLH transcription factors are widely distributed among eukaryotes and serve as activators or repressors of genes implicated in plant growth, development, and responses to environmental cues. These factors modulate subcellular localization, DNA-binding affinity, transcriptional activity, and protein stability [90]. In dandelion, the *TabHLH1* gene encodes a transcription factor that regulates the expression of key genes in the CGA biosynthetic pathway, including *TaHQT2*, *Ta4CL*, and *TaCHI*. Such regulation has been shown to significantly affect CGA concentrations [91].

The *ERF* gene family includes transcription factors that play crucial regulatory roles in a wide range of biological and physiological processes, including plant morphogenesis, stress response mechanisms, hormone signaling, and metabolic regulation [92]. In *N. tabacum*, the transcription factor encoded by the *NtERF4a* gene significantly upregulates the expression of *NtPAL1*, *NtPAL2*, *NtPAL3*, and *NtPAL4*, thereby promoting the accumulation of CGA in the leaves [93].

## 4. Promoting Chlorogenic Acid Synthesis—Microorganisms

### 4.1. Microorganisms: Unlock Plant Nutrient Uptake

Nutrients, as key drivers of plant secondary metabolism, directly modulate the diversity and abundance of secondary metabolites [11]. PGPBs are ubiquitous in soils, leaves, flowers, and plant tissues, where they improve nutrient uptake, growth, development, and secondary metabolism [94]. In CGA biosynthesis, PGPBs facilitate the uptake of essential nutrients, including nitrogen (N), phosphorus (P), potassium (K), magnesium (Mg), and iron (Fe) (Figure 3). This enhanced nutrient uptake promotes the synthesis of nucleic acids, enzymes, and hormones involved in CGA production. Moreover, PGPB activate critical metabolic pathways, such as photosynthesis, glycolysis, and the PPP, thereby augmenting CGA biosynthesis in plants.

N is a fundamental component of chlorophyll, proteins, nucleic acids, phospholipids, hormones, vitamins, and alkaloids, and it is essential for plant growth and development [95]. Chlorophyll, a key pigment for photosynthesis, critically influences plant growth and the synthesis of secondary metabolites, including CGA. Microorganisms facilitate N uptake in plants. For instance, leguminous plants release flavonoids from their roots into the soil, attracting rhizobia (Rh) to the rhizosphere and inducing the secretion of exopolysaccharides. These interactions lead to the formation of nodules that house rhizobial symbionts, which fix atmospheric N into ammonia for plant absorption [96]. Arbuscular mycorrhizal fungi (AMF), symbiotic partners of many plants, directly absorb NH_4_^+^ from the soil via extraradical hyphae and transfer it to plants. Alternatively, AMF can convert inorganic nitrogen (IN) into amino acids through glutamine synthetase and glutamate synthase in their extraradical mycelium. These amino acids are subsequently transformed into NH_4_^+^ by urease and urea amidolyase and provided to the host plant [97]. Furthermore, AMF can collaborate with soil bacteria and protozoa to acquire N from organic nitrogen sources, such as plant residues and chitin, transferring 20–75% of the total absorbed nitrogen to the host [98]. Inoculation of the *Trifolium repens* roots with Rh or AMF, either individually or in combination, was found to enhance N uptake in *T. repens*, with the combined inoculation yielding superior results compared to individual treatments [99,100].

P is essential for all terrestrial life and plays a critical role in photosynthesis, respiration, and the synthesis of nucleic acids and biological membranes [101]. Phosphate-solubilizing microorganisms (PSM) facilitate the conversion of unavailable soil inorganic phosphorus (Pi) into plant-accessible forms through two primary mechanisms: (i) acidification of the local environment via proton secretion, which dissolves inorganic phosphate minerals, and (ii) production of organic anions that chelate metal cations, dissolving minerals and releasing phosphorus as orthophosphate, thereby enhancing plant phosphorus uptake [99,102]. AMF can recruit bacteria carrying phosphatase genes and induce phosphatase production on their hyphal surfaces. These fungi then transport these bacteria to soil patches containing organic phosphorus (OP), increasing the efficiency of OP utilization and plant phosphorus uptake [103]. Co-inoculation with AMF and PSMs, compared to single inoculation, significantly enhances plant growth and phosphorus uptake while enhancing soil enzyme activity and rhizosphere microbial abundance [104]. For example, treating apple rootstocks with *B. B2* activates the transcription factor *MhMYB15*, increasing flavonoid accumulation and phosphorus uptake [105]. When tomato plants were inoculated with the nematophagus fungi *Pochonia chlamydosporia* and *Duddingtonia flagrans*, significant enhancements in plant growth and P uptake were observed compared to the control group [106].

K is a crucial macronutrient that regulates multiple physiological processes in plants, including stomatal opening and closing, photosynthesis, water uptake, hormone generation, maintenance of membrane potential, and sugar co-transport [107]. Additionally, K is required for the activation of over 60 enzymes [107]. Many rhizosphere microorganisms release organic and inorganic acids to dissolve mineral potassium (M-K) or produce extracellular polysaccharides, phenolic compounds, humic substances, and metal ions to acquire K through chemical reactions [108]. For example, inoculation of soybeans with *B. megaterium*, *Trichoderma longibrachiatum*, *T. simmonsii*, or a combination of these microorganisms enhances seed germination rates, seedling growth, and K uptake in the plants [109].

Magnesium ions (Mg²⁺) are essential cations in plant cells, serving as central binding ions for chlorophyll molecules to influence photosynthesis and acting as cofactors for over 300 enzymes that affect carbohydrate partitioning and protein synthesis [110]. Plant magnesium uptake is positively correlated with arbuscular mycorrhizal colonization [111]. For example, the inoculation of *T. aestivum* with *Glomus mosseae*, *Genus deserticola*, or *Gigaspora gergaria* increases leaf concentrations of chlorophyll a, chlorophyll b, and carotenoids, as well as tissue levels of P, N, K, Mg, and calcium [112].

Fe is a critical component of the plant electron transport chain and a cofactor for many essential enzymes, playing a vital role in photosynthesis and chlorophyll synthesis [113]. Iron availability in the soil influences crop yield and nutritional quality [114]. While iron is abundant in the lithosphere, its low solubility in water restricts the amount available for plant uptake, often failing to meet plant growth requirements [114]. Soil microorganisms synthesize and release specialized iron-binding compounds called siderophores [115]. These siderophores sequester iron from the environment and chelate ferric ions (Fe³⁺) to form Fe³⁺-siderophore complexes. A portion of these complexes is utilized by the microorganisms for growth, while another portion is absorbed by plants [116]. Thus, siderophores promote soil iron mobility and facilitate plant iron uptake. For example, siderophores produced by *P. fluorescens* strain C7R12 promote iron uptake in *A. thaliana* and upregulate approximately 2000 genes related to development and iron acquisition [117].

### 4.2. Microorganisms: Providers of Plant Hormones

Plant-associated microbes produce a diverse array of hormones, hormone-like substances, and compounds that modulate enzyme activities, thereby altering hormone levels in the plant’s endosphere, phyllosphere, and rhizosphere [118]. Hydroponic cultivation of GA-deficient rice and normal GA-biosynthesizing rice was conducted using the culture filtrate of *Paecilomyces formosus* LHL10. Seedling growth was significantly enhanced compared to the control group. Analysis of the culture filtrate identified the presence of multiple GAs (GA1, GA3, GA4, GA8, GA9, GA12, GA20, and GA24) and IAA [119]. ET synthesis can be regulated by the abundance or activity of ACC synthase or the availability of the ethylene precursor ACC [120]. Certain Rh (*Rhizobium*, *Sinorhizobium*, *Agrobacterium*, *Mesorhizobium*, and *Phyllobacterium* [121]) produce ACC deaminase, which breaks down ACC into ammonia and α-ketobutyrate, thereby inhibiting plant ethylene production. Studies have shown that the application of appropriate amounts of IAA and CK can increase plant dry matter yield, enhance stability, reduce disease incidence, and elevate phenolic compound content [122]. *Streptomyces* PM9 promotes root proliferation in *Eucalyptus grandis* by producing IAA and influencing its secondary metabolism [123]. Inoculation of tomato seedlings with *P. fluorescens* G20-18, which produces CK, enhanced the expression of key genes and enzyme activities involved in phenylalanine metabolism, carbohydrate metabolism, and antioxidant metabolism, ultimately increasing the levels of phenolics, flavonoids, and anthocyanins in tomato plants compared to a non-CK-producing mutant strain G20–18 [124]. Overall, microbes can produce a diverse array of plant hormones that promote the expression of enzymes, genes, and metabolic activities in plants, leading to the accumulation of phenolic and ester compounds (Figure 4).

### 4.3. Microorganisms: Gene Regulation

The composition, distribution, and structure of endophytic bacterial communities in plants are shaped by plant physiological characteristics, including genotype, morphological development, life history, and health status [125]. Consequently, microorganisms influence plant gene expression, morphological development, and health through complex interactions (Figure 4).

The progression and rate of CGA synthesis are regulated by enzymes, while microorganisms can modulate the expression of genes encoding CGA synthesis-related enzymes in plants, thereby controlling CGA biosynthesis. PAL is a key enzyme in this pathway. In *Cydonia oblonga* Mill, inoculated with *Rhizophagus intraradices* or *Funneliformis mosseae*, PAL activity and *PAL1* gene expression in seedling root systems are upregulated, along with a significant increase in phenolic compound content [126]. Similarly, *Mentha piperita* was treated with three PGPR strains—*P. fluorescens* WCS417, *P. putida* SJ04, and *B. subtilis* GB03—increases PAL activity and phenolic compound synthesis in leaves [127]. In studies on the defense of soybean against soybean cyst nematode using *B. simplex* strain Sneb545, Sneb545 suppressed two *PAL* genes while upregulating another, and both *4CL* genes in soybean were inhibited, resulting in the suppression of phenylpropanoid biosynthesis [128]. Additionally, *K. aerogenes* M2 was shown to promote the expression of aromatic-L-amino–acid/L-tryptophan decarboxylase, PAL, peroxidase, cinnamoyl-CoA reductase, shikimate *O*-hydroxycinnamoyl transferase, and *HCT* genes in *T. aestivum*, enhancing phenylalanine metabolism and the MAPK signaling pathway in *T. aestivum* roots [129].

Microorganisms regulate both the genes encoding plant enzymes and those of hormones that affect enzyme production and activity. For instance, inoculation of *Sedum alfredii* with *P. fluorescens* activates 146 genes involved in plant hormone signaling, increasing IAA concentrations and reducing ABA and JA levels in the roots [130]. In *T. aestivum* treated with *B. megaterium* N3, proteins related to DNA replication, plant hormones, and iron and zinc transport are upregulated in the roots [131]. Expression of ABA transporter genes *PYP/PYL* also increases, leading to elevated ABA content. Similarly, treatment of *Pennisetum glaucum* with *Shewanella putrefaciens* strain MCL-1 and *Cronobacter dublinensis* strain MKS-1 significantly increases ABA, IAA, and GA levels, along with upregulated expression of hormone biosynthesis-related genes *sbNCED*, *SbGA20oX*, and *SbYUC* [132]. By activating hormone-related genes, microorganisms enhance hormone production in plants, thereby influencing other physiological processes and metabolite accumulation.

Microorganisms regulate gene families encoding transcription factors, which are key elements in gene expression regulation. By interacting with specific DNA sequences, transcription factors modulate the expression of numerous genes, thereby influencing the biosynthesis and signaling of enzymes and hormones. For example, *B. velezensis* YYC promotes the expression of *WRKY33*, *WRKY22*, *ERF1*, *MYC2*, and 16 genes related to phenylpropanoid biosynthesis in *S. lycopersicum*, resulting in increased PAL activity in leaves and elevated levels of hormones such as SA and JA [133]. Under drought stress, co-inoculation of *Helianthus tuberosus* with *Rossellomorea aquimaris*, *Micrococcus luteus* strain 4.43, and *B. velezensis* strain 5.18 significantly upregulates the ethylene-responsive element *ERF1* gene in stems [134]. Additionally, *P. fluorescens* Ms9N and *Stenotrophomonas maltophilia* Ll4 can upregulate several genes from the *MYB* (*MtMYB74* and *MtMYB102*) and *WRKY* (*MtWRKY6*, *MtWRKY29*, *MtWRKY53*, and *MtWRKY70*) families in *Medicago truncatula* [135].

### 4.4. Microbiome: Interactions Between Bacteria, Fungi and Plants

Bacteria and fungi engage in interactions mediated by molecular mechanisms, including metabolite exchange, signaling, chemotaxis, physicochemical changes following adhesion, and protein secretion [136]. These interactions result in a spectrum of relationship types, ranging from antagonism to mutualism, and influence the growth, reproduction, motility, nutrition, stress resistance, and pathogenicity of both fungi and bacteria [136]. Moreover, these interactions play a crucial role in many agroecosystems, driving biogeochemical cycles and contributing to plant nutrition and health [137]. When these distinct microbial communities were transplanted into identical soils and used for cultivating *Bletilla striata*, both enhanced the growth of *B. striata* relative to a sterile soil control. Notably, the microbial community from sandy loam soil more effectively promoted the accumulation of militarine and other secondary metabolites in B. striata [138]. AM fungal communities in *Acer truncatum* Bunge from ten different regions revealed that superior trees harbored AM fungal communities with more complex symbiotic networks and specific taxa (VTX00069 and VTX00156), which positively influenced the accumulation of flavonoids and CGA in leaves [139]. Overall, the interactions between bacteria and fungi not only affect their own metabolism and reproduction but also significantly influence plant growth and metabolism.

## 5. Discussion and Prospects

(1)**Molecular mechanisms governing gene regulation**. Although initial understanding has been established regarding the myriad pathways that microbes employ to influence CGA biosynthesis in plants, the intricate molecular underpinnings of how these microbes exert control over plant gene expression are yet to be fully deciphered. To augment the repertoire of regulatory strategies aimed at enhancing plant CGA biosynthesis, it is imperative that forthcoming research endeavors delve into assessing the capacity of microbes to produce transcription factors or their homologs, alongside investigating alternative molecular pathways that may govern gene expression modulation.(2)**Gene Editing**. Current research has shown that microorganisms can regulate the expression of specific genes in plants, leading to increased CGA synthesis. Based on these findings, the CRISPR-Cas9 system can be employed to perform targeted modifications of these key genes in plants, thereby achieving efficient expression and high-yield synthesis of CGA.(3)**Microbial inocula and engineered microbial consortia**. Comprising beneficial microorganisms, microbial inocula present a viable strategy for soil health enhancement, serving as sustainable substitutes for conventional chemical fertilizers and pesticides [140]. Engineered microbial consortia, characterized by their simplified composition, high degree of control, and enhanced stability, represent a promising avenue for microbial manipulation [141]. Future research initiatives are poised to focus on CGA-rich plants, exemplified by species such as *E. ulmoides* and *L. japonica*. These studies will scrutinize the interplay between microorganisms and plants to pinpoint symbiotic or beneficial microbes that can be harnessed to augment CGA biosynthesis and accumulation within plants.(4)**Exogenous application of plant hormones**. It has been demonstrated that certain microorganisms, such as SA, JA, and MeJA, are capable of synthesizing and providing hormones to plants, which in turn promote the synthesis of CGA within plants. A strategy is proposed herein: engineering microorganisms to produce plant hormones [142] and applying them to plants at optimal concentrations to enhance plant growth, reproduction, CGA synthesis, and the production of other secondary metabolites. Moreover, this strategy can regulate the levels of plant hormones, optimize metabolic pathways in plants, and increase the yield of specific secondary metabolites.

## Figures and Tables

**Figure 1 microorganisms-13-01114-f001:**
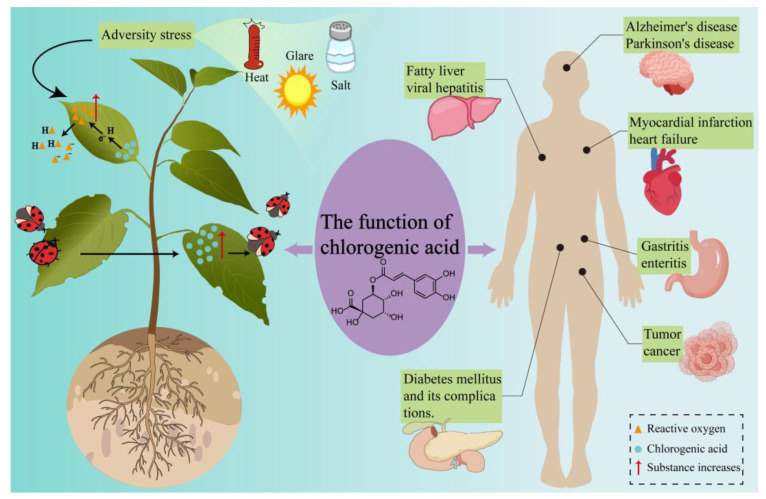
Biological and pharmacological functions of CGA. Yellow triangles—ROS; Blue circles—CGA; Red arrows indicate an increase in the respective substances.

**Figure 2 microorganisms-13-01114-f002:**
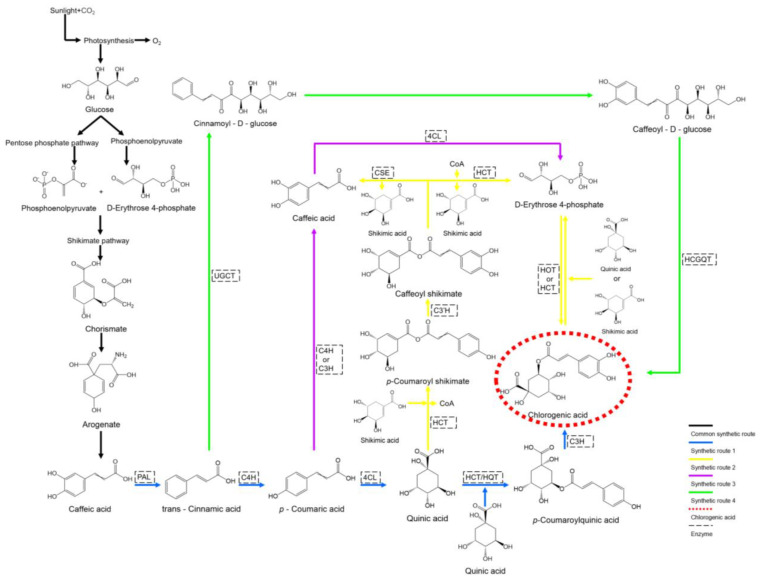
CGA biosynthetic pathways. The CGA biosynthetic pathways are numbered as 1, 2, 3, and 4. Black arrows—Common biosynthetic pathways; Blue arrows—CGA biosynthetic pathway 1; Yellow arrows—CGA biosynthetic pathway 2; Purple arrows—CGA biosynthetic pathway 3; Green arrows—CGA biosynthetic pathway 4; PAL—phenylalanine ammonia-lyase; C4H—cinnamate 4-hydroxylase; 4CL—4-coumarate-CoA ligase; HCT—hydroxycinnamoyl-CoA shikimate/quinate hydroxycinnamoyl transferase; HQT—hydroxycinnamoyl-CoA: quinate hydroxycinnamoyl transferase; C3H—*p*-coumarate 3-hydroxylase; C3′H—coumaroyl shikimate 3′-hydroxylase; UGCT—UDP glucose: cinnamate glucosyltransferase; HCGQT—hydroxycinnamoyl glucose: quinate hydroxycinnamoyl transferase; CSE—caffeoyl shikimate esterase.

**Figure 3 microorganisms-13-01114-f003:**
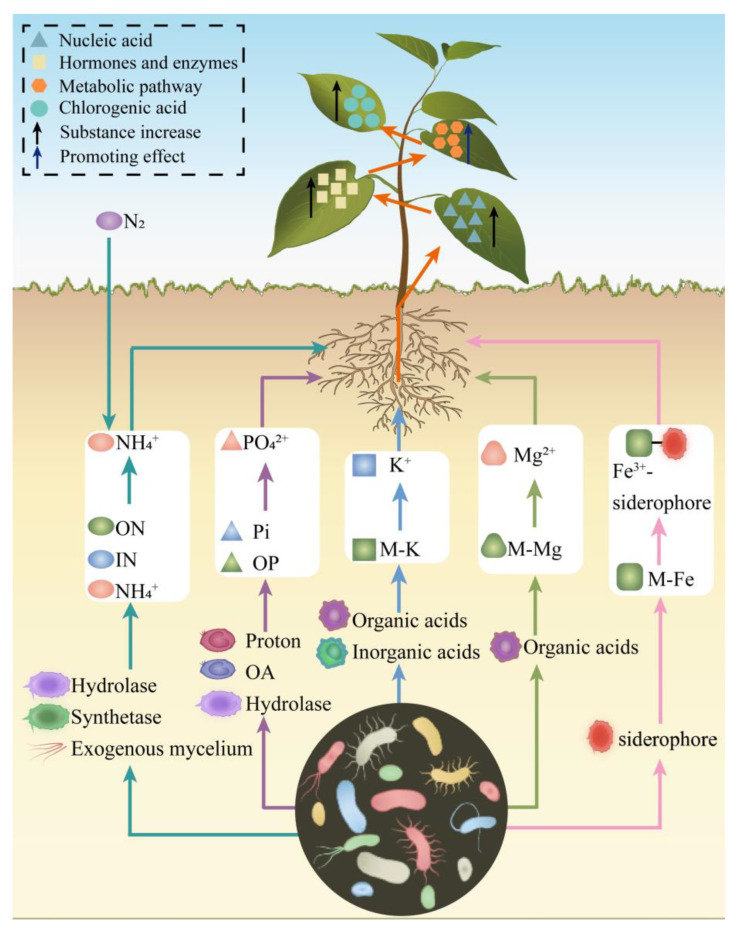
Microbial enhancement of plant nutrient acquisition. ON—Organic nitrogen; IN—Inorganic nitrogen; OA—Organic anions; OP—Organic phosphorus; Pi—Inorganic phosphorus; M-K—Mineral potassium; M-Mg—Mineral magnesium; M-Fe—Mineral iron; Orange arrows—Synergistic absorption of elements; Black arrows—Increase in substances; Dark blue arrows—Promotive effects; Blue triangles—Nucleic acids; Light yellow squares—Hormones and enzymes; Orange hexagons—Metabolic pathways; Blue circles—CGA.

**Figure 4 microorganisms-13-01114-f004:**
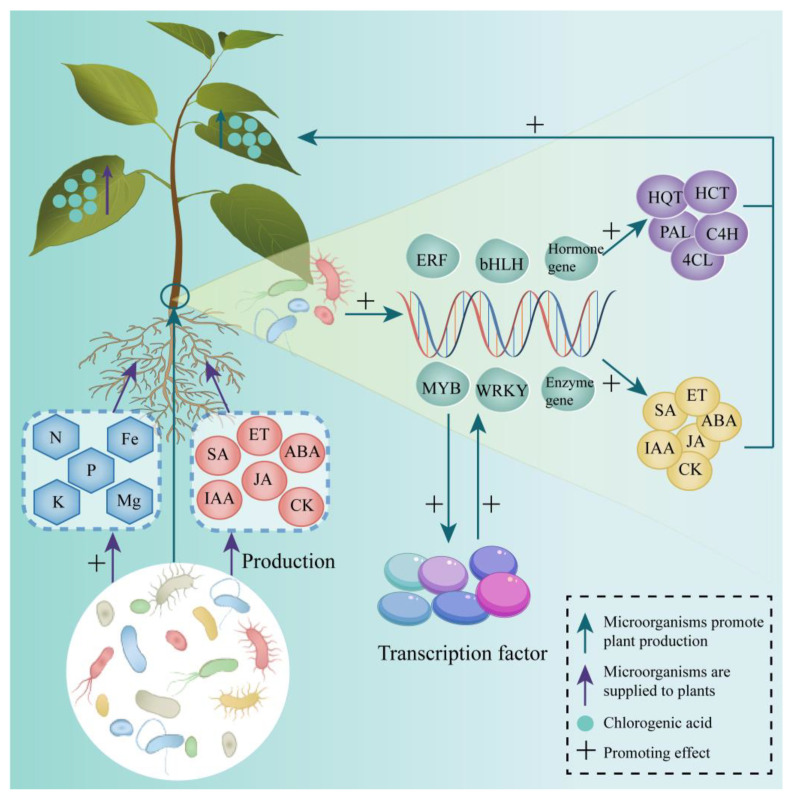
Mechanisms underlying microbial promotion of CGA Synthesis. Green arrows—Enhanced expression or production in plants mediated by microbes; Purple arrows—Substances provided by microbes to plants; Blue circles—CGA; +: Promoting effect.

## Data Availability

No new data were created or analyzed in this study.

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
