# Peer review of "Biosynthetic Mechanisms of Plant Chlorogenic Acid from a Microbiological Perspective"

_microorganisms, 2025, doi:10.3390/microorganisms13051114_

Round 1
Reviewer 1 Report
Comments and Suggestions for Authors
This review presents a presentation of the current understanding of chlorogenic acid (CGA) biosynthesis in plants, emphasizing microbial regulation. The manuscript is well-structured and extensively referenced, with clear explanation of the biosynthetic pathways, gene families, enzymes, and microbial roles in regulating CGA production. However, the manuscript would benefit from improvements in several key areas.
- Section 2.2: for the discussion of pharmacological features of chlorogenic acid in human body, their metabolism by the host/gut microbes should be briefly mentioned.
- The structures of caffeoyl shikimic acid and coumaroyl shikimate in Fig. 2 look wrong.
- Line 216. Be more specific about "activate". specify whether it's the enzyme activity or expression is activated.
- Line 235-244. The readers would benefit if the authors described the influence of hormones on CGA biosynthesis more quantitatively. Please include values.
- Line 247-252. The manuscript notes differing numbers of PAL genes across species but does not discuss why this occurs. A brief explanation of the evolutionary rationale and whether PAL isoforms have divergent functions or expression patterns within a species would benefit this discussion.
- Consider changing the section titles of 3.4.2 and 3.4.3 to better summarize the section contents. For example: 3.4.2: “Hormone Biosynthesis and Regulatory Genes”, 3.4.3: “Transcription Factors”
- Line 411-430: Evidence of these bacteria-produced metabolites are utilized by plants?
- Line 489. BGC is common for bacteria. But it seems not typical for plants to have clustered biosynthetic genes for secondary metabolite production.
- Missing citations: line 375, 384 and 392.
- p-coumaroyl (para), O-hydroxy (oxygen) need to be italicized where applicable.
Author Response
Comments 1: Section 2.2: for the discussion of pharmacological features of chlorogenic acid in human body, their metabolism by the host/gut microbes should be briefly mentioned.
Response 1: Based on your suggestions, the metabolism of chlorogenic acid in the host/gut microbiota has been added to the review. Please refer to lines 133-143.
Comments 2: The structures of caffeoyl shikimic acid and coumaroyl shikimate in Fig. 2 look wrong.
Response 2: Thank you for your careful inspection. The structure of coumaroyl shikimate in the figure is correct, but caffeoyl shikimic acid is written wrongly. It should generate caffeoyl shikimate. The content of the chart and text has been corrected. Please see lines 171 and 185.
The references are: Barros J, Escamilla-Trevino L, Song L, et al. 4-Coumarate 3-hydroxylase in the lignin biosynthesis pathway is a cytosolic ascorbate peroxidase[J]. Nature Communications, 2019, 10(1): 1994.
Comments 3: Line 216. Be more specific about "activate". specify whether it's the enzyme activity or expression is activated.
Response 3: I have read the original literature and corrected the inaccurate expression here. It has been corrected to "the activity of the activating enzyme". Please refer to line 227.
Comments 4: Line 235-244. The readers would benefit if the authors described the influence of hormones on CGA biosynthesis more quantitatively. Please include values.
Response 4: Thank you for your detailed comments. We have supplemented the relevant detailed data on the impact of hormones on CGA synthesis. Please refer to lines 245-256.
Comments 5: Line 247-252. The manuscript notes differing numbers of PAL genes across species but does not discuss why this occurs. A brief explanation of the evolutionary rationale and whether PAL isoforms have divergent functions or expression patterns within a species would benefit this discussion.
Response 5: Thank you for your suggestions on this part. We deleted this part based on the opinions of the second reviewer and added information such as the distribution of the PAL gene. Please refer to Articles 262-270.
Comments 6: Consider changing the section titles of 3.4.2 and 3.4.3 to better summarize the section contents. For example: 3.4.2: “Hormone Biosynthesis and Regulatory Genes”, 3.4.3: “Transcription Factors”
Response 6: We think this is a very good suggestion. We have modified the title. Please see lines 309 and 324.
Comments 7: Line 411-430: Evidence of these bacteria-produced metabolites are utilized by plants?
Response 7: Thank you for your question on this. We have added examples here that can confirm that the metabolites produced by bacteria can be utilized by plants. Please refer to lines 438-442 and 449-455.
Comments 8: Line 489. BGC is common for bacteria. But it seems not typical for plants to have clustered biosynthetic genes for secondary metabolite production.
Response 8: Thank you for your opinion. We have reviewed the relevant literature and found that BGC is indeed atypical in plants. Therefore, we have decided to delete this outlook.
Comments 9: Missing citations: line 375, 384 and 392.
Response 9: Thank you for pointing out the omissions in the citations of these three references. By comparing the contents of these three places, we found that they all come from the same reference as the previous or subsequent sentence. We have already made sentence modifications and added references. Please refer to lines 401, 411 and 418.
Comments 10: p-coumaroyl (para), O-hydroxy (oxygen) need to be italicized where applicable.
Response 10: We are sorry for our carelessness. Thank you for your reminder. We have italicized "p" and "O".

Reviewer 2 Report
Comments and Suggestions for Authors
Comments to manuscript” Biosynthetic Mechanisms of Plant Chlorogenic Acid from a Microbiological Perspective”, submitted by Zhong et al.
Although there are some similarities to the article of Wang et al (2922), the review of Zhong et al. is interesting as it presents the function of the compound in plant protection against pathogens and other stresses, recent insight in the biosynthesis of chlorogenic acid in plants including genes and enzymes, the role of plant hormones and signal transduction in regulating the biosynthesis and others. The title of the review points to the involvement of microorganisms in promoting the biosynthesis by facilitating nutrient uptake by the plant, by bacterial hormone production and the interplay with the plant, the influence of endophytic bacteria on plant gene expression, here of those with functions in the biosynthesis of chlorogenic acid /other phenolics. In the last part, some considerations are presented for the use of innovative methods to optimize plant-microbe interactions and to elucidate microbial strategies to modulate plant gene expressions.
The major concern is the almost exclusive focus on bacterial microorganisms. Except for Trichoderma longibrachiatum and some arbuscular mycorrhizal fungi, the latter are omitted. This is particularly incomprehensible as fungi may produce chlorogenic acid, for instance, in the plant Eucommia ulmoides – several times mentioned in the manuscript- Sordariomycete sp. strain B5 was identified as a chlorogenic acid producer. Also, fungi are known to degrade chlorogenic acid, for instance Fusarium graminearum or Aspergillus niger. Chlorogenic acid is converted to protocatechuic acid, an intermediate which is further transformed to compounds able to enter the citrate cycle. Moreover, fungi are known to produce plant hormones as well. Thus, there are interactions that involve fungi, bacteria and the plant.
Another point are the number of PAL and other genes mentioned in the manuscript: “potatoes possess more than 40” (ref. 63): Check the reference. The work of Han et al. deals with Iridaceae, and not with Solanaceae. "The apple genome contains 69 4CL genes [69], while E. ulmoides harbors 35 Eu4CL genes [47]": Concerning the apple 4CL genes, the genes “were obtained by homology analysis” and “and there were great differences in the number of amino acids encoded by different genes even in the same subfamily” (wrote Ma et al.). I think this result must be verified and it is the question if this study should be mentioned in a review. It is possible that most of the gene products have no PAL activity. Enzyme assays were not performed. Concerning E. ulmoides, the proteins encoded by “the 35 sequences with the 4CL conserved domain” may not have 4CL activity. And what about the fungal endophytes?
The authors should include aspects related to fungi in their review. They cannot exclude this important group of organisms. The restriction only to bacteria (in case the few mentioned fungi are deleted in a revised manuscript), will give a wrong picture.
Author Response
Comments 1: The major concern is the almost exclusive focus on bacterial microorganisms. Except for Trichoderma longibrachiatum and some arbuscular mycorrhizal fungi, the latter are omitted. This is particularly incomprehensible as fungi may produce chlorogenic acid, for instance, in the plant Eucommia ulmoides – several times mentioned in the manuscript- Sordariomycete sp. strain B5 was identified as a chlorogenic acid producer. Also, fungi are known to degrade chlorogenic acid, for instance Fusarium graminearum or Aspergillus niger. Chlorogenic acid is converted to protocatechuic acid, an intermediate which is further transformed to compounds able to enter the citrate cycle. Moreover, fungi are known to produce plant hormones as well. Thus, there are interactions that involve fungi, bacteria and the plant.
Response 1: Sincerely thank you for these opinions you put forward. In fact, our topic is "Biosynthetic Mechanisms of Plant Chlorogenic Acid from a Microbiological Perspective". It focuses on how microorganisms promote the production of chlorogenic acid in plants. Regarding your mention that we overlooked the role of fungi, we have made modifications and added an example of fungi in microorganisms promoting the absorption of nutrients. Please refer to lines 378-380 and 395-398. Examples of fungi have also been added in the section on the production of hormones by microorganisms. Please refer to sections 438-442. Regarding the issue you raised about the interaction among bacteria, fungi and plants, we have added a section on "The interaction between Bacteria, Fungi and Plants" in the section on microorganisms promoting chlorogenic acid. Please refer to lines 503-520.
Comments 2: Another point are the number of PAL and other genes mentioned in the manuscript: “potatoes possess more than 40” (ref. 63): Check the reference. The work of Han et al. deals with Iridaceae, and not with Solanaceae. "The apple genome contains 69 4CL genes [69], while E. ulmoides harbors 35 Eu4CL genes [47]": Concerning the apple 4CL genes, the genes “were obtained by homology analysis” and “and there were great differences in the number of amino acids encoded by different genes even in the same subfamily” (wrote Ma et al.). I think this result must be verified and it is the question if this study should be mentioned in a review. It is possible that most of the gene products have no PAL activity. Enzyme assays were not performed. Concerning E. ulmoides, the proteins encoded by “the 35 sequences with the 4CL conserved domain” may not have 4CL activity. And what about the fungal endophytes?
Response 2: Thank you for your detailed suggestions. We believe that your statement that the enzymes produced by the PAL, 4CL and C4H genes may not all have enzymatic activity is correct. Therefore, we have rewritten this part of the content, deleted the number of genes corresponding to the same enzyme in different plants, and added examples that can prove the activity of the enzyme produced by gene expression. Please see lines 261-295.
Comments 3: The authors should include aspects related to fungi in their review. They cannot exclude this important group of organisms. The restriction only to bacteria (in case the few mentioned fungi are deleted in a revised manuscript), will give a wrong picture.
Response 3: Thank you for your opinion. We have added the content about fungi promoting chlorogenic acid in the article, as well as bacteria,
For the interaction between fungi and plants, please refer to lines 503-520. Meanwhile, in the outlook section, we also mentioned the synthetic microbial community, which also includes fungi.

Round 2
Reviewer 2 Report
Comments and Suggestions for Authors
Dear authors,
the suggestions have been considered in the revised manuscript. I have no further concerns.